# The Influence of Time Winning and Time Losing on Position-Specific Match Physical Demands in the Top One Spanish Soccer League

**DOI:** 10.3390/s21206843

**Published:** 2021-10-14

**Authors:** José C. Ponce-Bordón, Jesús Díaz-García, Miguel A. López-Gajardo, David Lobo-Triviño, Roberto López del Campo, Ricardo Resta, Tomás García-Calvo

**Affiliations:** 1Faculty of Sport Sciences, University of Extremadura, Boulevard of the University s/n, 10003 Caceres, Spain; jponcebo@gmail.com (J.C.P.-B.); malopezgajardo@unex.es (M.A.L.-G.); davidlobo123@gmail.com (D.L.-T.); tgarciac@unex.es (T.G.-C.); 2LaLiga Sport Research Section, 28043 Madrid, Spain; rlopez@laliga.es (R.L.d.C.); rresta@laliga.es (R.R.)

**Keywords:** contextual variables, match running performance, ball possession, positional, professional soccer

## Abstract

The aim of the present study was to analyze the influence of time winning and time losing on position-specific match physical demands with and without ball possession in the top Spanish professional soccer league. All matches played in the First Spanish soccer league over four consecutive seasons (from 2015/16 to 2018/19) were recorded using an optical tracking system (i.e., ChyronHego), and the data were analyzed via Mediacoach^®^. Total distance (TD), and TD > 21 km·h^−1^ covered with and without ball possession were analyzed using a Linear Mixed Model, taking into account the contextual variables time winning and losing. Results showed that TD and TD > 21 km·h^−1^ covered by central midfielders (0.01 and 0.005 m/min, respectively), wide midfielders (0.02 and 0.01 m/min, respectively), and forwards (0.03 and 0.02 m/min, respectively) significantly increased while winning (*p* < 0.05). By contrast, TD and TD > 21 km·h^−1^ covered by central defenders (0.01 and 0.008 m/min, respectively) and wide defenders (0.06 and 0.008 m/min, respectively) significantly increased while losing (*p* < 0.05). In addition, for each minute that teams were winning, total distance with ball possession (TDWP) decreased, while, for each minute that teams were losing, TDWP increased. Instead, TDWP > 21 km·h^−1^ obtained opposite results. Total distance without ball possession increased when teams were winning, and decreased when teams were losing. Therefore, the evolution of scoreline significantly influences tactical–technical and physical demands on soccer matches.

## 1. Introduction

Context-related variables are considered the most influencing variables on match physical demands in soccer [1]. Time–motion analysis research has reported a large amount of information about context-related variables such as match status, match location, and opponent level [2]. Specifically, it has been previously shown that final and partial match status (analyzed by epochs of time—i.e., 15-min periods or half-time) modify match physical demands as well as ball possession [3,4]. However, this method of analysis is probably to do with the interaction of other variables such as the evolution of the match-scoreline [5]. Regardless of final match status, the time each team was leading, drawing, or trailing during a match could be different, with matches taking place where a team has won throughout 70 min (i.e., team A scored a goal in the 20th min and team B did not score) or matches where a team has won throughout 1 min (i.e., teams were drawing over the match, and one team scored a goal in the 89th min). Moreover, it is possible that one team that was ahead for a long time would end up losing on the final scoreline. Match physical demands are believed to depend on evolving scoreline (i.e., whether a team is winning or losing) since, when a team is losing, players try to reach their maximal physical capacity in order to draw or win the match [2]. Therefore, this limitation could be solved by taking into account the minutes that teams were ahead and behind during a match separately. According to our knowledge, the influence of time winning and time losing on position-specific match physical demands according to the evolution of the scoreline is less known.

Match status has been arguably analyzed enough to prove that it influences soccer teams’ behavior [6]. In this vein, Lago–Peñas [7] reported that losing teams frequently increase their percentage of possession; meanwhile, certain winning teams preferred counterattacking or playing directly. In addition, match status clearly impacts teams playing style [5], and both variables (i.e., match status and playing style) also influence match physical demands; however, several studies about this topic have drawn the opposite conclusions. For instance, elite Spanish soccer players performed less high-intensity distance (19 km·h^−1^) when winning than when they were losing, since winning is a comfortable status, it is possible that players assume a ball contention strategy, keeping the game slower, which results in lower speeds [8]. Moreover, Castellano et al. [2] analyzed one Spanish team of LaLiga and they showed that the distances covered at high intensity by the reference team were greater when the result was adverse. Moalla et al. [3] obtained the same results in a study from the Stars League during 2013/14 and 2014/15 seasons. Conversely, during qualifying round matches of the World Cup 2010, drawing teams covered a lower average speed than the winning and losing teams [9]. In this line, variables that determine the intensity of the game (maximum speed and frequency of high-intensity activity) in a professional Brazilian football team were significantly lower when the team lost [10], meanwhile greater intensity running distances were observed in matches that the team won as opposed to losing [11]. Therefore, coaches should take into account the match status to analyze the external load implied by the match [12].

However, soccer players could have not been affected by this contextual variable in the same way due to playing style, since playing style changes associated with match status could affect different players differently in ways that are position-specific [13]. For example, in the German Bundesliga during the 2014/15 season, central defenders and full-backs covered shorter distances at high intensity in won matches than in lost matches (*p* < 0.01); however, forwards covered significantly longer total distance in won matches than in drawn and lost matches (*p* < 0.05) [14]. Despite these conclusions, position-specific match physical demands can also vary depending on the evolution of scoreline status. In this vein, when a team was winning, during preseason matches of the 2011/12 season Australian League soccer, the average speed was 4.17% lower than when the team was drawing (*p* < 0.05) [15]. Regarding player position-specific data, in the English Premier League, Redwood-Brown et al. [16] found midfielders covered more distance at high intensity when level, defenders more when losing, and attackers more when winning. Similarly, losing status increased the total distance covered by defenders from Spanish First soccer league, while attacking players showed the opposite trend [17]. Therefore, these previous studies have suggested that, due to the player position, players perform different tactical roles, and match status and the associated playing style changes could have different influences on match running performance by positions.

The knowledge about the influence of time that teams were ahead or behind on position-specific match physical demands could have important practical applications during the competitive season to program the training load in a more strategic way based on physical data [12]. In addition, less is known about the influence of time which teams spend winning or losing on position-specific match physical demands. Therefore, the main objective of the present study was to analyze the influence of time winning and losing on position-specific match physical demands in the top Spanish soccer league across four seasons (2015/2016–2018/2019). As a secondary objective, the study also aimed to analyze the match physical demands with and without ball possession according to time winning and time losing.

Based on previous findings obtained by the aforementioned studies, the following hypotheses were proposed. Firstly, concerning match physical demands with and without ball possession, it was expected that total distance with ball possession would be less during time winning [7]. Secondly, concerning position-specific match physical demands, it was expected that total distance would be greater in attackers during time winning and defenders during time losing, based on previous results [16,17].

## 2. Materials and Methods

### 2.1. Participants

The sample comprised 36,883 individual match observations of 1037 professional soccer players who competed in the First Spanish professional soccer league (i.e., LaLiga Santander) over four consecutive seasons (from 2015/16 to 2018/19). All players who participated in matches (starters and non-starters) and played 10 min at least were included. Only goalkeepers were excluded. According to previous studies [18], players were divided into five position-specific groups: Central Defenders (CD; *n* = 6787 observations), Wide Defenders (WD; *n* = 6530 observations); Central Midfielders (CM; *n* = 6826 observations); Wide Midfielders (WM; *n* = 8394 observations); Forwards (FW; *n* = 8346 observations). Data were provided to the authors by LaLigaTM, and the study received ethical approval from the University of Extremadura; Vice-Rectorate of Research, Transfer and Innovation-Delegation of the Bioethics and Biosafety Commission (Protocol number: 239/2019).

### 2.2. Procedure and Variables

Match physical demands data were collected by an optical tracking system (ChyronHego^®^; TRACAB, New York, NY, USA). This multi-camera tracking system consists of 8 different super 4K-High Dynamic Range cameras situated strategically to follow and track the 22 players on the field throughout the match. These cameras film from several angles and analyze X and Y coordinates of each player, providing real-time tracking with data recorded at 25 Hz. Mediacoach^®^ is also based on data correction of the semi-automatic video technology (the manual part of the process) [19]. The validity and reliability of the Tracab^®^ video tracking system has been analyzed, reporting average measurement errors of 2% for total distance covered [20,21,22].

The physical performance variables used for this study were categorized according to the ball possession as follows [23,24]: with possession (WP) and without possession (WOP). The following variables were studied for each of these categories: total distance (m) covered by players (i.e., TD) and total distance covered at more than 21 km·h^−1^ (i.e., TD > 21 km·h^−1^).

To determine if the scoreline influenced position-specific match physical demands, the cumulative time that each team was losing or winning during a match was included in the analysis (not the final match result). For example, if team A scored a goal in the 20th min and team B equalized in the final minute, team A was classified as losing for 0 min and winning for 70 min, while team B was classified as losing for 70 min and winning for 0 min

### 2.3. Data Analysis

All statistical analyses were performed using R-studio [25]. A Linear Mixed Model (LMM) was conducted for each of the physical variables using the lme4 package [26]. This model allows for the analyzing of data with a hierarchical structure in nested units and has demonstrated its ability to cope with unbalanced and repeated-measures data [27]. For example, variables related to the distance covered in matches are nested for players (i.e., each player has a record for every match they have participated in), and players are nested into teams. Also, cumulative times spent winning or losing are nested into matches and these matches can also be nested into teams. This represents a threefold levels structure, where teams are the topmost unit in the hierarchy.

A general multilevel-modelling strategy was applied [27], where fixed and random effects had been included in different steps from the simplest to the most complex. First, unconditional models were analyzed exclusively including dependent variables (i.e., distance variables) to check if the grouping variables at levels 2 and 3 (i.e., players and teams) significantly affected the intercept (mean) of each dependent variable. These models may be used as baselines for comparing more complex models. Later, different models were performed for each of the dependent variables, setting as fixed effects the position of the players and the time winning/losing. Following the procedure proposed by Heck & Thomas [27], models with different random effects (intercepts and slope) were created for each variable. A model comparison for each step was performed using the Akaike Information Criterion (AIC) [28] and a chi-square likelihood ratio test [29], where a lower value represented a fitter model. Final models presented in Table 1 and Table 2 (with random intercept and slope effect) were chosen according to better values of AIC, log-likelihood, and significant effect of variables. Maximum Likelihood (ML) estimation for model comparison and for the final model of each physical variable was used, the best model, again, using Restricted Maximum Likelihood (REML) estimation, was refitted. Marginal and conditional R^2^ metrics [30] for each LMM to provide some measure of effect sizes were reported. Significance level was set at *p* < 0.05.

For a suitable interpretation of the results, the time winning/losing was group-mean centered, being centered to the team’s mean in each season.

## 3. Results

Firstly, the Wald test and intraclass correlation coefficient (ICC) suggested statistically significant variability in the distances covered by players according to time winning and losing (ICC > 0.10); therefore, LMM was justified for the purpose of the study. Also, AIC suggested that the twofold levels model was the one fitter for this purpose.

Secondly, Table 1 shows the differences of TD covered according to ball possession and to the scoreline evolution by player positions. Regardless of scoreline, CM covered significantly greater TD than the rest of the players (*p* < 0.05). TD covered by CD and WD decreased significantly with respect to CM, WM, and FW (*p* < 0.05) for each minute that teams were ahead. By contrast, for each minute that teams were trailing, TD covered by CD and WD increased significantly with respect to CM, WM, and FW (*p* < 0.05).

During the match, FW covered significantly greater TDWP than CD, WD, CM, and WM (*p* < 0.05). However, for each minute that teams were ahead, TDWP decreased for all positions. Significant differences were found between WD and CM with respect to FW (*p* < 0.01). On the contrary, for each minute that teams were trailing, TDWP increased for all positions. Significant differences between CM and FW were observed (*p* < 0.05).

On the other hand, CM covered TDWOP significantly greater than the rest of the players (*p* < 0.05). However, for each minute that teams were ahead, TDWOP increased for all positions. CM and WM significantly increased TDWOP with respect to CD, WD, and FW (*p* < 0.01). By contrast, for each minute that teams were trailing, TDWOP significantly decreased for all positions, except CD (*p* < 0.05).

Thirdly, Table 2 shows the differences by ball possession of TD > 21 km·h^−1^ according to the scoreline evolution by player positions. Regardless of the scoreline, FW covered TD > 21 km·h^−1^ significantly greater than the rest of the players (*p* < 0.05). However, for each minute that teams were ahead, CD and WD significantly decreased TD > 21 km·h^−1^ with respect to CM, WM, and FW (*p* < 0.05). By contrast, for each minute that teams were trailing, CD and WD significantly increased TD > 21 km·h^−1^ with respect to CM, WM, and FW (*p* < 0.05).

During the match, FW covered TD > 21 km·h^−1^ with ball possession significantly greater than CD, WD, CM, and WM (*p* < 0.05). Moreover, for each minute that teams were ahead, TD > 21 km·h^−1^ significantly increased for all positions, except WD (*p* < 0.05). By contrast, for each minute that teams were trailing, TD > 21 km·h^−1^ significantly decreased for all positions, except WD (*p* < 0.05).

Finally, WD covered significantly greater TD > 21 km·h^−1^ without ball possession than the rest of the players (*p* < 0.01). For each minute that teams were ahead, TD > 21 km·h^−1^ without ball possession covered by WM and FW increased significantly, with respect to CD, WD, and CM (*p* < 0.05). Likewise, for each minute that teams were trailing, WD and CM significantly increased TD > 21 km·h^−1^ without ball possession, with respect to CD, WD and FW (*p* < 0.05).

## 4. Discussion

The aim of the present study was to analyze the influence of time winning and losing on position-specific match physical demands in the top Spanish professional soccer league across four seasons (from 2015/2016 to 2018/2019). Subsequently, match physical demands with and without ball possession according to time winning and losing were examined. The main results showed that TDWP was less while teams were winning, while it was greater while teams were losing. In addition, TDWOP increased while teams were winning, while it decreased while teams were losing. Finally, TD and TD > 21 km·h^−1^ covered by CM, WD, and FW were greater while teams were winning, while TD and TD > 21 km·h^−1^ covered by CD and WD were greater while teams were losing.

Firstly, it had been hypothesized that TDWP would be less during time winning (*Hypothesis 1*). Our results showed that TDWP decreased during time winning for all player positions and increased during time losing, therefore Hypothesis 1 was confirmed. These results suggest that during time winning, teams frequently decrease their percentage of possession, which could be associated to defending closer to the goal, counterattacking, or playing directly [7]. It has also been shown that teams that were ahead performed a higher number of defensive actions which, in turn, are related to lower ball possession levels during a match [31]. By contrast, during time losing, these results suggest teams frequently increase their percentage of possession, attacking closer to the other team’s goal [7]. Evidence has reported that successful teams normally have longer possession times than less successful teams [23] or that ball possession might increase in teams that are either losing or trying to tie the match [32]. Therefore, our results reported the need to take into account the evolution of the scoreline.

In contrast, TDWP >21 km·h^−1^ increased during time winning for all player positions and decreased during time losing. A possible reason to explain this result could be the fact that teams adopt an indirect playing style to perform counterattacks [7]. Therefore, players need to execute high-intensity specific technical and tactical tasks on the pitch when they are in ball possession, such as receiving passes and crosses on the run, followed by dribbling the ball in the opponent’s area to obtain a goal [33]. In fact, it has been demonstrated that high-intensity actions are important within decisive situations in professional football [34]. Furthermore, Yang et al. [24] reported that total sprint distance was significantly greater for the best-ranked teams compared to lower-ranked teams, highlighting the importance of sprinting for tactical teamwork that generates offensive actions. A systematic review conducted by Lago–Peñas & Sanromán–Álvarez [35] pointed out that successful teams covered greater high-intensity running distance in ball possession. Therefore, our results suggest that teams perform a greater number of high-intensity actions in ball possession while winning or trying to maintain the advantage; meanwhile, TDWP decreases.

During time winning, TDWOP increased for all player positions and decreased during time losing for all player positions, except CD. One potential reason for this situation could be the fact of ball possession decrease, like Lago-Peñas [7] reported, showing that winning teams preferred counterattacking or playing directly. On the other hand, research has shown that lower-ranked teams covered significantly greater TDWOP compared with better-ranked teams, which likely represented a greater match time undertaking defensive activities by these teams [24]. Our findings disagree with those from other studies where ball possession increased when teams were ahead [36]. Similarly, TDWOP > 21 km·h^−1^ covered by WM and FW significantly increased when teams were ahead, and TDWOP > 21 km·h^−1^ covered by WD and CM increased when teams were behind. This fact could be explained due to that, when the team is not in possession of the ball, the forwards often perform high-intensity activities (high pressing), attempting to recover the lost ball [14,37].

Secondly, it had been hypothesized that TD would be greater in attackers while teams were ahead and in defenders when teams were losing (*Hypothesis 2).* The results showed that TD and TD > 21 km·h^−1^ covered by CM, WM, and FW significantly increased when teams were ahead (*p* < 0.05), and TD and TD > 21 km·h^−1^ covered by CD and WD significantly increased when teams were losing (*p* < 0.05). These findings are in line with previous studies that found that attackers covered more distance at high intensity when winning and defenders more when losing [16]. Lago–Peñas et al. [17] also reported that losing status increased total distance covered by defenders, while attacking players showed the opposite trend. Thus, it seems to be confirmed that CM, WM, and FW cover greater TD when winning and CD and WD when losing, therefore Hypothesis 2 was accepted. A possible explanation may be due to attackers´ work rate of the opposing team since, when the opposing team is ahead or chasing a goal, attackers maintain a high work rate, implying a defenders´ high work rate [16,38]. In addition, similar results were obtained by Andrzejewski et al. [14], showing that defenders covered shorter distances at high intensity in lost matches, while forwards covered longer total distances in won matches. Another possible reason could be the playing style that the team adopted, for example, a direct style of play when teams are winning can induce higher match intensity in running from the attackers [7,11]. In particular, these findings indicate that physical demands vary according to position-specificities and the evolution of the scoreline.

### 4.1. Study Limitations and Future Directions

The present study increases the knowledge about this research topic; however, a number of limitations could be recognized with a view to future research. First, other context-related variables such as match location or opposing team level were not considered. Ball possession percentages of teams have been also not considered, and it would be interesting to analyze this variable when teams are winning or losing with the interaction of match physical demands. Moreover, the comparison between five players’ positions according to previous studies was analyzed [18]; however, the existence of more player positions is possible than have been previously analyzed, therefore it would be interesting to conduct a comparison between more player positions. In addition, further research is required, considering several factors such as the playing style, since the player position could depend on the playing style of teams. Finally, research has reported that external load variables such as accelerations and decelerations belong to match physical demands [39], in which case it would be necessary to know the full player´s work rate, including these variables.

### 4.2. Practical Applications

The findings of this study provide useful information on the variability of match physical demands for practitioners in Spanish professional soccer. In particular, the study extends previous research demonstrating that time that teams were winning or losing influences both match physical demands and ball possession. This information could help strength and conditioning coaches with personalizing recovery work after match play, according to the different physical efforts performed in matches. Finally, goals scored are the most important of all critical events, therefore the evolution of the scoreline should be taken into account during training sessions to optimize physical aspects of soccer performance. In this vein, it is necessary to know how these situations influence the player’s capacity to deal with critical events in a match [40].

## 5. Conclusions

The main findings reported that the evolution of scoreline significantly influences match tactical–technical and physical demands. First, TDWP was less while teams were winning, while it was greater while teams were losing, and TDWOP evolved conversely; therefore, teams modify their playing style and tactical behavior according to the demands of matches. Secondly, attackers covered greater distances when winning, and defenders covered greater distances when losing; therefore, professional soccer players regulate their physical efforts according to the periods of the game. Finally, the influence of scoreline is reflected in changes in the teams and players’ tactical–technical and physical demands as a response to the evolution of match outcome.

## Figures and Tables

**Table 1 sensors-21-06843-t001:** Differences by ball possession of position-specific total distance covered according to the scoreline evolution.

		CD	WD	CM	WM	FW
TD (m/min)	Intercept	107.30 b, c, d, e	109.90 a, c, d, e	116.10 a, b	115.90 a, b	115.60 a, b, c
	Slope Time Winning	−0.006 c, d, e	−0.005 c, d, e	0.01 a, b, d, e	0.02 a, b, c, e	0.03 a, b, c, d
	Slope Time Lossing	0.01 c, d, e	0.006 c, d, e	−0.003 a, b, d, e	−0.02 a, b, c, e	−0.02 a, b, c, d
TDWP (m/min)	Intercept	35.93 b, c, d, e	38.41 a, c, d, e	42.21 a, b, e	42.54 a, b, e	43.04 a, b, c, d
	Slope Time Winning	−0.03	−0.04 e	−0.04 e	−0.04	−0.03 b, c
	Slope Time Lossing	0.03	0.03	0.03 e	0.02	0.02 c
TDWOP (m/min)	Intercept	42.50 b, c, d	43.80 a, c, d, e	47.39 a, b, d, e	45.36 a, b, c, e	42.16 b, c, d
	Slope Time Winning	0.01 c, d, e	0.02 d, e	0.03 a, d, e	0.04 a, b, c	0.01 a, b, c
	Slope Time Lossing	0.002 d, e	−0.003 d, e	−0.003 d, e	−0.01 a, b, c	−0.02 a, b, c

Note. CD = Central defenders; WD = Wide defenders; CM = Central midfielders; WM = Wide midfielders; FW = Forwards; TD = Total distance; TDWP = Total distance with ball possession; TDWOP = Total distance without ball possession; a = significant differences compared to central defenders; b = significant differences compared to wide defenders; c = significant differences compared to central midfielders; d = significant differences compared to wide midfielders; e = significant differences compared to forwards.

**Table 2 sensors-21-06843-t002:** Differences by ball possession of position-specific total distance covered at more than 21 km·h^−1^ according to the scoreline evolution.

		CD	WD	CM	WM	FW
Total distance > 21 km·h^−1^ (m/min)	Intercept	5.74 b, c, d, e	6.68 a, d, e	6.68 a, d, e	7.24 a, b, c, e	7.57 a, b, c, d
	Slope Time Winning	−0.008 c, d, e	−0.005 c, d, e	0.005 a, b, d, e	0.01 a, b, c, e	0.02 a, b, c, d
	Slope Time Lossing	0.008 c, d, e	0.008 c, d, e	−0.004 a, b, d, e	−0.001 a, b, c, e	−0.01 a, b, c, d
Total distance with ball possession > 21 km·h^−1^ (m/min)	Intercept	2.21 b, c, d, e	2.81 a, c, d, e	3.14 a, b, d, e	3.57 a, b, c, e	4.01 a, b, c, d
	Slope Time Winning	0.001 c, d, e	−0.001 c, d, e	0.005 a, b, e	0.007 a, b, e	0.01 a, b, c, d
	Slope Time Lossing	−0.001 a, c, d, e	0.001 a, b, d, e	−0.006 a, b, c	−0.009 a, b, c	−0.009 a, c, d, e
Total distance without ball possession > 21 km·h^−1^ (m/min)	Intercept	3.54 b, c, d, e	3.80 a, c, d, e	3.37 a, b, e	3.40 a, b, e	3.16 a, b, c, d
	Slope Time Winning	−0.008 b, c, d, e	−0.005 a, c, d, e	−0.001 a, b, d, e	0.002 a, b, c	0.004 a, b, c
	Slope Time Lossing	−0.009 b, c, d, e	0.007 a, c, d, e	0.002 a, b, d, e	−0.001 a, b, c, e	−0.002 a, b, c

Notes. CD = Central defenders; WD = Wide defenders; CM = Central midfielders; WM = Wide midfielders; FW = Forwards; a = significant differences compared to central defenders; b = significant differences compared to wide defenders; c = significant differences compared to central midfielders; d = significant differences compared to wide midfielders; e = significant differences compared to forwards.

## Data Availability

Restrictions apply to the availability of these data. Data was obtained from LaLiga and are available with the permission of corresponding author.

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
