# Peer review of "The Influence of Time Winning and Time Losing on Position-Specific Match Physical Demands in the Top One Spanish Soccer League"

_sensors, 2021, doi:10.3390/s21206843_

Round 1

Reviewer 1 Report

Hello.
The article falls both in the area of the field and continues research in this direction and in a sport in the forefront of the public.
The presentation of the research is well documented, scientifically substantiated and complies with the latest standards regarding the highest level scientific publications.
A strong point I consider to be the approvals and the procedure used, the use of the technology and the data it provides as well as the correlation of the results and their interpretation.
The conclusions support and result from the research carried out and open new directions for future research.
Congratulations on the method found to scientifically demonstrate how the score influences the physical commitment of football players.
Congratulations. All the best!.

Reviewer 2 Report

The influence of time winning and time losing 16 on position-specific match physical demands with and without ball possession in the top Spanish 17 professional soccer league was analysed in this study.  It was found that the total distance without 27 ball possession increased when teams were winning, while, decreased when teams were losing. 28, the evolution of scoreline significantly influences tactical-technical and physical de- 29
mands on soccer matches. Very novel finding and impressive work!

Reviewer 3 Report

The article "The influence of time winning and time losing on position-specific match physical demands in the top one Spanish soccer league" is interesting. Congrats to the authors of the idea. I find small editorial errors. See my comments.

L80 "(p < . 05)" change on "(p < . 05)".

Incorrect form of reference:
L 396 / L399 / L421 - bold the year and add pages.

Thank you for the opportunity to review.

Reviewer 4 Report

Dear Authors,

in the manuscript entitled "The influence of time winning and time losing on position specific match physical demands in the top one Spanish soccer league" by José Carlos Ponce-Bordón and his coworkers, the Authors carried out a detailed analysis of all matches played in the First Spanish soccer league over four consecutive seasons. I recognize the efforts Authors have put into this work. The sample of 36 883 individual match observations is a big amount of data. The Authors proposed two research hypotheses that were verified on the basis of the obtained results and statistical analyzes. Result of research and main conclusion presented by the Authors has practical application and can be useful for managing a soccer team  as well as establishing a training plan.

After carefully going through the manuscript, although I found the work is solid in the field of statistics and processing of large amounts of data, yet, its findings look more appropriate for other journals like Stats (2571-905X) rather than Sensors. The main objection comes from the fact that the whole study has marginal connections with the scope of Sensors ("Sensors (ISSN 1424-8220) provides an advanced forum for the science and technology of sensor and its applications."). I would suggest the authors to change some parts of the article (acording to the remark) and transfer the manuscript to Stats journal.

Specific remarks/editorial comments/typos:

  • Line 7-15, Why the same affiliation is repeated 5 times and the second twice?
  • The Abstract section should present quantitative results and not only the most important qualitative results and/or generic considerations.  In my opinion the specific research results contained in the article are missing. Therefore, significant improvements are expected.
  • In line 99, 104, 105 and others, the Authors use the personal form ("…we also aimed ..."). This is not correct in high-quality articles. It suggests modifying this part of the article. Please use passive verb or different verb in active voice. Please check the entire text of the manuscript.
  • Line 122-127 - Auyhors describe the optical tracking system used during the research. Why is there no system diagram and view of the system as well as view of the data recorded by the system? The description of the system itself is very general. The article does not contain drawings or other graphic objects.
  • The Authors did not describe the measuring system in detail in the article. What is new in this measurement system that distinguishes it from others available on the market or described in the scientific literature?
  • The article contains a number of abbreviations, I recommend adding a list of abbreviations at the end of the article along with a detailed explanation of each, for example TD - total distance covered by players. The article will be more readable for the reader.
  • Line 190 - the table caption should be centered.
  • Table 1 and 2 - I suggest adding units to the individual parameters presented in the tables.
  • In chapter 4.1 Study limitations and future directions, Authors have written "In addition, further research is required considering several factors such as the playing style, since the player position could depend on the playing style of teams." Why do the Authors do not take into account the number of minutes played by a player/team? There are teams in the Spanish Soccer League that play every 3 days (i.e. Champions League), while others only have league matches. The resulting difference may have an impact on the physical aspect.

The article requires the above changes. I hope these suggestions can help to improve the quality of this paper.

I wish you all the best.

Round 2

Reviewer 4 Report

Dear Authors,

thank you for taking into account the comments described in the first version of the review. The Authors comments as well as modyfication of the article and the information about the Special issue convinced me. The article may be published in the Sensors journal.

I wish you all the best